# Metabolic profiles among COPD and controls in the CanCOLD population-based cohort

**Damien Viglino[1,2]\*, Mickaël Martin[1], Marie-Eve Piché[1], Cynthia Brouillard[1], Jean-Pierre Després[1], Natalie Alméras[1], Wan C. Tan[3], Valérie Coats[1], Jean Bourbeau[4], Jean-Louis Pépin[2], François Maltais[1], on behalf of the CanCOLD Collaborative Research Group and the Canadian Respiratory Research Network[¶]**

**1** Centre de Recherche, Institut Universitaire de Cardiologie et de Pneumologie de Québec, Université Laval, Québec, Canada, **2** Centre Hospitalier Universitaire de Grenoble-Alpes, Laboratoire HP2 INSERM U 1042 Université Grenoble-Alpes, Grenoble, France, **3** James Hogg Research Centre, University of British Columbia, Vancouver, Canada, **4** Montreal Chest Hospital, Mc Gill University, Montreal, Canada

¶ Membership of the CanCOLD Collaborative Research Group and the Canadian Respiratory Research Network is provided in the Acknowledgments.

\* damien.viglino@criucpq.ulaval.ca

**Data Availability Statement:** All relevant data are within the manuscript and its Supporting Information files.

## Abstract

A high prevalence of intermediate cardiometabolic risk factors and obesity in chronic obstructive pulmonary disease (COPD) has suggested the existence of pathophysiological links between hypertriglyceridemia, insulin resistance, visceral adiposity, and hypoxia or impaired pulmonary function. However, whether COPD contributes independently to the development of these cardiometabolic risk factors remains unclear. Our objective was to compare ectopic fat and metabolic profiles among representative individuals with COPD and control subjects and to evaluate whether the presence of COPD alters the metabolic risk profile. Study participants were randomly selected from the general population and prospectively classified as non-COPD controls and COPD, according to the Global Initiative for Chronic Obstructive Lung Disease classification. The metabolic phenotype, which consisted of visceral adipose tissue area, metabolic markers including homeostasis model assessment of insulin resistance (HOMA-IR), and blood lipid profile, was obtained in 144 subjects with COPD and 119 non-COPD controls. The metabolic phenotype was similar in COPD and controls. The odds ratios for having pathologic values for HOMA-IR, lipids and visceral adipose tissue area were similar in individuals with COPD and control subjects in multivariate analyses that took into account age, sex, body mass index, tobacco status and current medications. In a population-based cohort, no difference was found in the metabolic phenotype, including visceral adipose tissue accumulation, between COPD and controls. Discrepancies between the present and previous studies as to whether or not COPD is a risk factor for metabolic abnormalities could be related to differences in COPD phenotype or disease severity of the study populations.

**Funding:** The Canadian Cohort Obstructive Lung Disease (CanCOLD) study is funded by the Canadian Respiratory Research Network (CRRN); industry partners: Astra Zeneca Canada Ltd; Boehringer Ingelheim Canada Ltd; GlaxoSmithKline Canada Ltd; and Novartis. Researchers at RI-MUHC Montreal and Icapture Centre Vancouver lead the project. Previous funding partners are the CIHR (CIHR/Rx&D Collaborative Research Program Operating Grants 93326); the Respiratory Health Network of the Fonds de la recherche en santé du Québec (FRSQ); industry partners: Almirall; Merck Nycomed; Pfizer Canada Ltd; and Theratechnologies. The funders had no role in study design, data collection and analysis, decision to publish, or preparation of the manuscript. Additional research support grants were provided to DV by "Agir Pour les Maladies Chroniques" Foundation and "Fond de Recherche Québec-Santé", both non-profit organisations.

**Competing interests:** MM, CB, NA and VC have no conflicts of interest to declare. DV report grants from Astra Zeneca France outside the submitted work. MEP is research scholars from the Fonds de Recherche du Québec-Santé (FRQ-S) JPD reports personal fees from Abbott Laboratories, AstraZeneca, GSK, Merck and Pfizer Canada Inc. and personal fees from Abbott Laboratories, Sanofi and Torrent Pharmaceuticals Ltd. outside the submitted work. WCT reports grants from Canadian Institute of Heath Research (CIHR/Rx&D Collaborative Research Program Operating Grants-93326) with industry partners AstraZeneca Canada Ltd, Boehringer Ingelheim Canada Ltd, GlaxoSmithKline Canada Ltd, Merck, Novartis Pharma Canada Inc., Nycomed Canada Inc., Pfizer Canada Ltd, during the conduct of the study. JB reports grants from Canadian Institute of Heath Research (CIHR/Rx&D Collaborative Research Program Operating Grants-93326) with industry partners AstraZeneca Canada Ltd, Boehringer Ingelheim Canada Ltd, GlaxoSmithKline Canada Ltd, Merck, NovartisPharma Canada Inc., Nycomed Canada Inc., Pfizer Canada Ltd, during the conduct of the study. JLP report grants from Air Liquide Foundation, Agiradom, AstraZeneca, Fisher and Paykel, Mutualia, Philips, Resmed and Vitalaire outside the submitted work and personnal fees from Agiradom, AstraZeneca, Boehringer Ingelheim, Jazz pharmaceutical, Night Balance, Philips, Resmed and Sefam outside the submitted work. FM reports grants and personal fees from Boehringer Ingelheim and GSK, grants from Nycomed and grants and personal fees from Novartis outside the submitted work. All fees are pooled with other revenues of the group of pulmonologists to which FM is a member and then

## Introduction

Cardiometabolic diseases are at the forefront of comorbidities in the Chronic Obstructive Pulmonary Disease (COPD) population [1]. It has been reported that individuals with COPD have a 2- to 5-time higher risk of cardiovascular disease compared with controls, independently of shared risk factors such as age and smoking [2,3]. Understanding the nature of the link between COPD and co-existing metabolic conditions/comorbidities may provide personalized treatment strategies and identify new mechanistic pathways to be targeted.

The relationship between COPD and its comorbidities is complex and studies having reported a high prevalence of metabolic syndrome and obesity in patients with COPD [4–7] have suggested the existence of pathophysiological links between hypertriglyceridemia and hypoxia [8,9], obesity and hypoxia [10–12], or visceral adiposity and pulmonary function [13–16]. Various phenotypes of COPD have emerged, some of which being defined by the adiposity and metabolic profile of the patients [17–20]. In a previous investigation [21], we found that the degree of visceral adiposity with its associated hypertension and diabetes correlated with the severity of COPD [Global initiative for Obstructive Lung Disease (GOLD) grade]. Several potential confounders (tobacco exposure, dietary habits, sedentarity) may, however, complicate the establishment of a link between COPD and metabolic abnormalities.

In the present investigation based on the above-mentioned cohort, we aimed to further explore whether COPD is linked to established metabolic variables (insulin resistance [22–26], lipid control [27] and visceral adiposity [28,29]) in a well-phenotyped cohort representative of the general population. We hypothesized that if there is causal and self-sustaining links between COPD and metabolic abnormalities, then differences in metabolic risk factors should emerge between individuals with COPD and control subjects. The present study was embedded in the Canadian Cohort Obstructive Lung Disease Study (CanCOLD), a prospective longitudinal study of COPD with random population sampling [30].

## Methods

### Participants

The study was approved by the local ethics committee (Comité d'éthique du centre de recherche de l'Institut Universitaire de Cardiologie et de Pneumologie de Québec, IRB N° 20690, Study N° 2012–1359). CanCOLD (ClinicalTrials.gov: NCT00920348) steering and scientific committees approved the sub-study protocol. All study participants signed written consent before inclusion.

Participants in two CanCOLD study centres (Montreal and Quebec City, Quebec, Canada) were recruited between February 2012 and December 2015 for this sub-study. CanCOLD is a longitudinal cohort study based on the characterization of COPD among a random sample of the population in 9 Canadian cities [30]. Subjects had to be 40 years or older to participate in the CanCOLD study. Further details concerning the CanCOLD study design and eligibility criteria have been previously described [30]. Study participants underwent the standard CanCOLD assessment procedures, which provide information about patients' characteristics (age, gender, smoking history), medical history and current medications, body weight and height, and pulmonary function. Although no sleep studies were done in CanCOLD, the presence of sleep apnea was documented based on the use of continuous airway positive pressure (CPAP) and on standardized questionnaires, including the Pittsburg Sleep Quality Index [31]. Additional pre-specified measures were done including measurements of waist and hip circumferences, blood sampling to determine glucose and lipid profiles, and a computed tomography (CT) abdominal scan at $4^{th}/5^{th}$ lumbar vertebrae level (L4-L5) to quantify body fat distribution

shared among members of the group. This does not alter our adherence to PLOS ONE policies on sharing data and materials.

[21]. Participants were divided according to the pulmonary function testing results as follows: 1) control subjects with a post-bronchodilator forced expiratory volume in 1 second ($FEV_1$) > 80% predicted value and $FEV_1$/forced vital capacity (FVC) ratio > 0.7; 2) patients with COPD with a post-bronchodilator $FEV_1$/FVC ratio < 0.7 were further classified according to the Global Initiative for Chronic Obstructive Lung Disease (GOLD) airflow limitation classification scheme into GOLD 1, with an $FEV_1 \geq 80\%$ predicted, and GOLD 2+ with an $FEV_1 < 80\%$ predicted. All COPD subjects were invited to be enrolled in the final CanCOLD cohort, whereas some of the healthy subjects were enrolled to serve as controls with a control/COPD ratio of 1 to 1 [30]. Patients with a pulmonary restrictive profile were excluded from the analysis.

## Procedures

**Body fat distribution and visceral adipose tissue assessment.** L4-L5 CT scan images were analyzed without knowledge of the clinical status of the subjects. Abdominal fat distribution was assessed using the specialized software Tomovision SliceOMatic (v4.3 Rev-6f, Montreal, Quebec, Canada). The detailed method used for image analysis has been previously reported [32, 33]. The middle of the muscle wall surrounding the abdominal cavity was delineated to determine the visceral adipose tissue (VAT) area. Abdominal adipose tissue areas were computed using an attenuation range of –190 to –30 Hounsfield units (HU). Body fat distribution parameters were obtained with methodology commonly applied in our Core Lab, with high levels of intra and inter-observer agreement [32].

**Blood sample and biochemical analysis.** Blood samples were collected in the morning, after a 12-hour fast to determine levels of glucose, insulin, total cholesterol, LDL-cholesterol, HDL-cholesterol and triglycerides. All analyses were carried-out in plasma or whole blood using automated techniques (Roche Diagnostics). Glucose, total cholesterol (TC), HDL-cholesterol, LDL-cholesterol, and triglycerides were measured by enzymatic *in vitro* tests. Insulin was determined using electrochemiluminescence immunoassay (ECLIA). Insulin resistance was assessed using the homeostatic model assessment for insulin resistance (HOMA-IR), calculated using the following formula: insulinemia × glucose/22.5 (glucose units mmol/L) [34].

## Data analysis

Continuous data are presented as median and interquartile range (IQR) or mean and 95% confidence interval in case of normal distribution, and categorical data as frequency and percentage. Continuous variables were analysed using a Mann-Whitney test and categorical data and proportions were analysed using the Fisher exact tests. Metabolic phenotypes were compared between COPD and controls by using four complementary strategies: 1) univariate comparisons of adiposity and metabolic parameters (triglycerides, total/HDL cholesterol ratio, and HOMA-IR) between COPD subjects, GOLD 1 subjects, GOLD 2+ subjects and controls (Mann-Whitney test); 2) univariate linear regression with coefficient of determination ($R^2$) and analysis of covariance (ANCOVA) to study the relationships between metabolic parameters (triglycerides, total/HDL cholesterol ratio, HOMA-IR) and indices of adiposity (body mass index (BMI), waist-to-hip ratio and VAT area) according to COPD status; 3) multivariate linear regression models to detect possible interactions between the COPD status and the various metabolic parameters studied. A logarithmic transformation (Ln) was performed on each non-log-linear variable of interest. These models took into consideration (variable entry) all potential confounders available, including age, sex, smoking status, BMI, waist-to-hip ratio, corticosteroid treatment, hypolipidemic and hypoglycemic agents. Final models were selected with backwards elimination, with COPD status as a forced variable and keeping only the

significant variables at $p <0.05$; and 4) multivariate logistic regressions to estimate the odds ratio of having hypertriglyceridemia (triglyceride >1.5 mmol/L), increased total /HDL cholesterol ratio>4 [35], and insulin resistance (HOMA-IR>3) [22–26] in the presence of COPD (all COPD and COPD GOLD2+ only) compared to non-COPD controls. These models were adjusted for potential confounders (age, sex, smoking status, BMI, corticosteroid treatment and ongoing pharmacological treatment related to the parameter studied, namely hypolipidemic drugs or hypoglycemic drugs). The odds ratio of having visceral obesity (L4-L5 VAT cross-sectional area >75[th] percentile of the whole population by sex) in the presence of COPD (all COPD and COPD GOLD2+ only) in comparison to non-COPD controls was analysed by multivariate logistic regression including age, smoking status and inhaled corticosteroid treatment as known confounding factors. In these multivariate logistic regressions, continuous variables were entered as quartiles. Missing data were not replaced. All statistical analyses were performed using IBM SPSS v.23 software (IBM statistics, USA) and GraphPad Prism v6.05 (GraphPad Software, USA).

## Results

This CanCOLD sub-study included 263 participants having a median age of 65 [59–71] years and of whom two thirds were males. Based on pulmonary lung function, subjects were divided into control subjects with normal lung function (n = 119), and individuals with COPD (n = 144, 70 GOLD 1 and 74 GOLD 2+). No missing data in variables of interest have to be reported. There was no statistically significant between-group difference for age, sex, BMI, waist-to-hip ratio, and use of hypolipidemic and oral hypoglycemic agents (**Table 1**).

Metabolic profiles according to COPD status are provided in **Fig 1**. There was no significant difference between groups in triglyceride levels (**Fig 1A**), total/HDL cholesterol ratio (**Fig 1B**), and insulin resistance (HOMA-IR) (**Fig 1C**). The median VAT levels in control subjects (146.4 cm$^2$ [106.2–222.6]) was not different compared to COPD subjects (155.7 cm$^2$ [108.9–233.9], $p$ = 0.59), and to GOLD 1 or GOLD 2+ COPD (**Fig 1D**). No significant difference was observed between COPD and controls in univariate analysis stratified by BMI for any metabolic parameter (**S1 Fig**).

Triglycerides, total/HDL cholesterol ratio and HOMA-IR were positively associated with the three indices of adiposity (BMI, waist-to-hip ratio and VAT area) in individuals with COPD and controls. (**Fig 2**, all regression lines with a p<0.05). However, the slopes of the regression lines were similar for both groups (p>0.05 for all comparisons) suggesting that the relationships between metabolic markers and adiposity were not modified in the presence of COPD.

In linear multivariate analyses, the COPD status was not significantly associated with triglyceride levels, total/HDL cholesterol, insulin resistance (HOMA-IR) or VAT area (**Table 2**). A higher BMI was associated with an increase in triglycerides, total/HDL cholesterol, and HOMA-IR levels, and current smokers with one third of additional VAT. No significant interaction was observed between the COPD status and any characteristic tested.

Lastly, a triglyceride level above 1.5 mmol/L, a total/HDL cholesterol ratio above 4 or a HOMA-IR above 3 were respectively observed in 54 (37.5%), 36 (25%) and 58 (40.3%) of COPD patients and in 37 (31.1%), 33 (27.7%) and 54 (45.4%) of control subjects. In multivariate analysis, the COPD status (or COPD 2+) was not associated with triglyceride >1.5 mmol/L, total/HDL cholesterol ratio >4, HOMA-IR>3, or VAT area>75[th] percentile. Only the COPD 2+ status was associated with a VAT area>75[th] percentile (OR = 2.27, CI95% 1.00; 5.15, p = 0.05). Complete regression models are available in supplementary S1–S8 Tables.

**Table 1. Baseline characteristics by group.**

|  | Control subjects (n = 119) | COPD (n = 144) | P value |
|---|---|---|---|
| Age, years | 65 [59–71] | 65 [59–71] | 0.88 |
| Male, n (%) | 73 (61.3) | 93 (64.6) | 0.61 |
| BMI, kg/m$^2$ | 26.5 [23.5–29.7] | 26.6 [23.7–29.4] | 0.96 |
| Waist-to-hip ratio, mean (95% CI) | 0.93 (0.92–0.94) | 0.94 (0.93–0.95) | 0.14 |
| Waist circumference, cm | 96 [87.8–103] | 98 [89–106] | 0.20 |
| Current smokers, n (%) | 12 (10.1) | 39 (27.1) | <0.001 |
| Former smokers, n (%) | 70 (58.8) | 74 (51.4) | 0.26 |
| Never smokers, n (%) | 37 (31.1) | 31 (21.5) | 0.09 |
| Pack/year | 11 [0–28] | 27 [0–50] | <0.001 |
| **Comorbidities** |  |  |  |
| Hypertension, n (%) | 32 (26.9) | 53 (36.8) | 0.11 |
| Diabetes, n (%) | 10 (8.4) | 14 (9.7) | 0.83 |
| Dyslipidemia, n (%) | 31 (26.1) | 41 (28.5) | 0.68 |
| Coronary artery disease, n (%) | 6 (5.0) | 14 (9.7) | 0.17 |
| Stroke, n (%) | 1 (0.8) | 9 (6.3) | 0.02 |
| Sleep apnea, n (%) | 4 (3.4) | 9 (6.3) | 0.39 |
| **Pulmonary Function, post BD** |  |  |  |
| FEV$_1$, L | 2.88 [2.37–3.48] | 2.14 [1.55–2.99] | <0.001 |
| FEV$_1$, % predicted | 101 [92–110] | 79 [65–93] | <0.001 |
| FVC, L | 3.79 [3.09–4.52] | 3.63 [2.75–4.69] | 0.48 |
| FVC, % predicted | 120 [112–132] | 118 [103–135] | 0.21 |
| FEV$_1$/FVC, % | 76.8 [73.4–79.9] | 62.3 [55.7–66.5] | <0.001 |
| PEF, mean L/sec (95% CI) | 7.42 (6.98–7.86) | 7.42 (7.02–7.82) | <0.001 |
| FEF 25–75, L/sec | 1.60 [0.96–2.41] | 1.58 [0.95–2.41] | <0.001 |
| GOLD 1, n (%) | - | 70 (48.6) | - |
| GOLD 2, n (%) | - | 61 (42.4) | - |
| GOLD 3–4, n (%) | - | 13 (9.0) | - |
| GOLD A, n (%) | - | 91 (63.2) | - |
| GOLD B, n (%) | - | 40 (27.8) | - |
| GOLD C, n (%) | - | 3 (2.1) | - |
| GOLD D, n (%) | - | 10 (6.9) | - |
| **Medications at baseline** |  |  |  |
| Short-acting BD, n (%) | 2 (1.7) | 24 (16.6) | <0.001 |
| Long-acting BD, n (%) | 1 (0.8) | 1 (0.7) | 1 |
| Inhaled CS, n (%) | 3 (0.3) | 33 (22.9) | <0.001 |
| Statins, n (%) | 28 (23.5) | 38 (26.4) | 0.67 |
| Other hypolipidemic drugs, n (%) | 4 (3.3) | 3 (2.1) | 0.70 |
| Insulin, n (%) | 1 (0.8) | 0 (0) | 0.45 |
| Oral hypoglycemic agents, n (%) | 9 (7.6) | 10 (6.9) | 1 |

Values are median [IQR] if not stated otherwise. COPD: chronic obstructive pulmonary disease; BMI: body mass index; CI: confidence interval; BD: bronchodilator; FEV$_1$: forced expiratory volume in 1 second; FVC: forced vital capacity; BD: bronchodilator; CS: corticosteroids; GOLD: global Initiative for obstructive lung disease classification.

## Discussion

In a population-based cohort consisting of individuals with mild to moderate COPD and control subjects, we found metabolic profiles (lipid profile, HOMA-IR, and VAT accumulation)

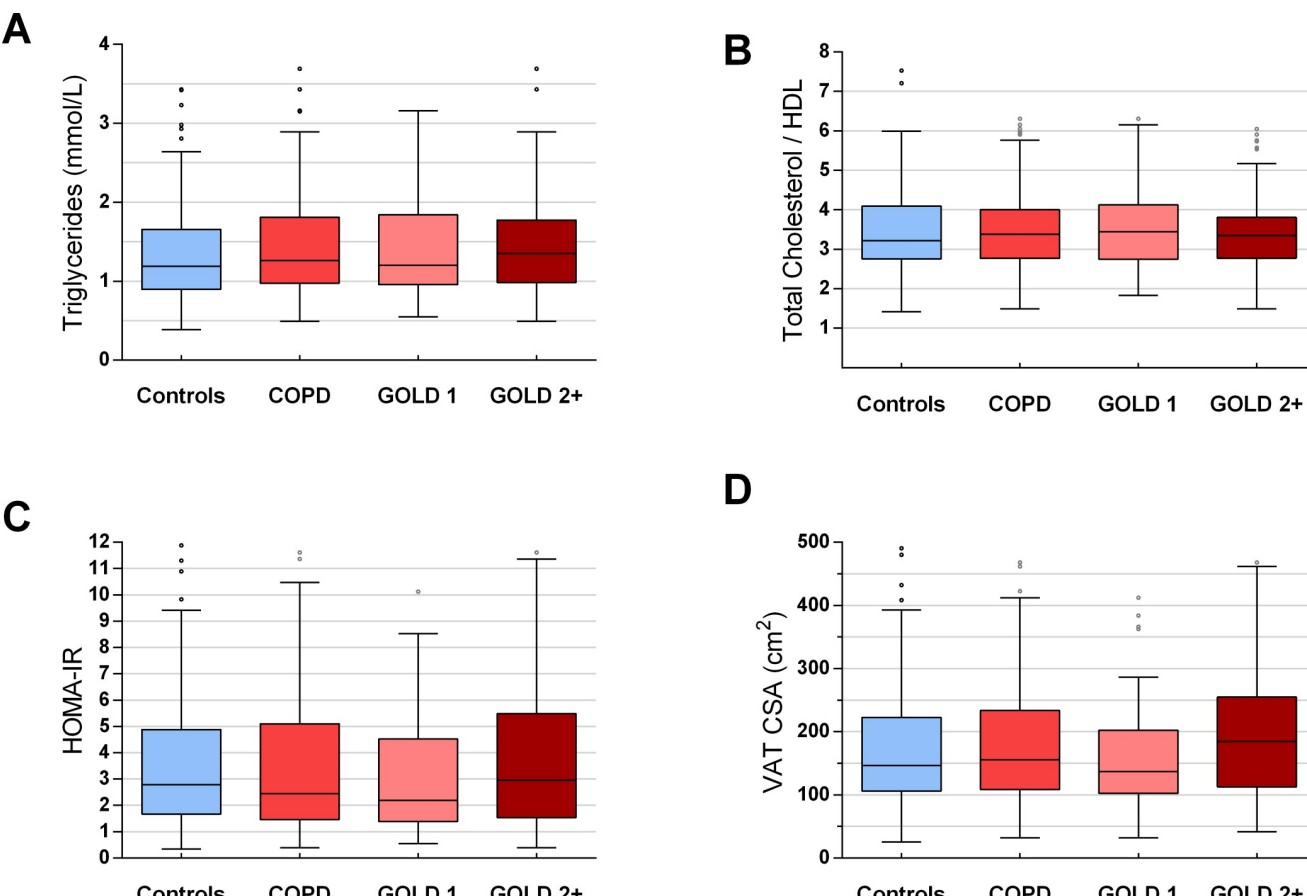

**Fig 1. Metabolic parameters according COPD status.** COPD: chronic obstructive pulmonary disease; HDL: high density lipoprotein; HOMA-IR: homeostasis model assessment of insulin resistance; VAT CSA: visceral adipose tissue cross-sectional area on L4-L5. p>0.05 for all between-group comparisons.

that were not influenced by the presence of COPD. The well-established relationships between triglycerides, total/HDL cholesterol ratio, and HOMA-IR to indices of adiposity [36,37], which were confirmed here, were not modified in the presence of COPD. Univariate and multivariate analyses showed an absence of association between COPD and metabolic disorders or visceral adiposity. Therefore, based on this thorough statistical approach, we conclude that COPD does not emerge as an independent risk factor for metabolic disorders and visceral adiposity in a cohort that can be considered representative of the entire population.

Numerous studies have explored possible physiopathological links between COPD, asthma or sleep apnea and cardiometabolic components [38]. In those respiratory diseases, several bidirectional mechanisms have been proposed to enhance the risk of hypertriglyceridemia, adipose tissue accumulation and insulin resistance, including hypoxia [8–12] and hypercapnia [39]. Activation of lipolysis in adipose tissue in the presence of hypoxia led to the "adipose tissue hypoxia" concept [11]. Adipose tissue would then appear to play a central role in the development of chronic inflammation, macrophage infiltration, and would also be responsible for increasing circulating free fatty acids [8,10,11]. In addition, fat-induced systemic inflammation involving adipokines [38,40–42], insulin and its receptor, has been implicated in lung injury and airway responsiveness [38,43,44], causing a deleterious pathophysiological loop.

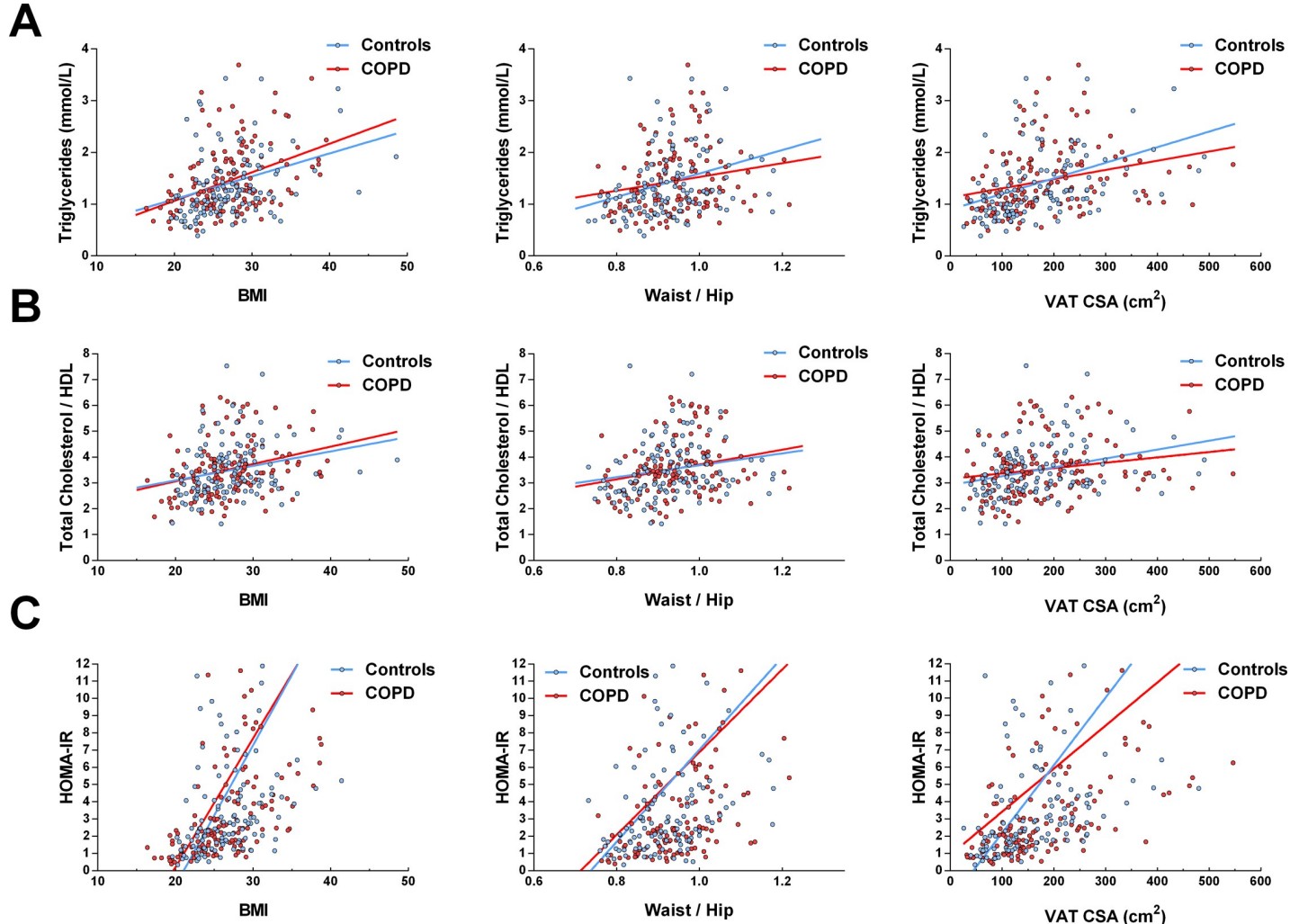

**Fig 2. Relationships between metabolic parameters and BMI, waist-to-hip ratio and VAT CSA in individuals with COPD and controls.** COPD: chronic obstructive pulmonary disease; BMI: body mass index; HDL: high density lipoprotein; HOMA-IR: homeostasis model assessment of insulin resistance; VAT CSA: visceral adipose tissue cross-sectionnal area on L4-L5. All coefficients of determination ($R^2$) are<0.3; All regression line slopes were significantly different from 0 ($p<0.05$); however, none of the regression lines couples (COPD vs. controls) were significantly different ($p>0.05$ for all comparisons).

In light of the above potential pathophysiological links between chronic respiratory diseases and cardiometabolic risk factors, it was deemed legitimate to propose that COPD may contribute to the development of metabolic abnormalities. In one of the most large-scale studies in the field, Leone et al. [13] found an association between lung function impairment and "classical" components of the metabolic syndrome. This result was obtained in a heterogeneous population (obstructive and restrictive ventilatory defects), and a sub-analysis restricted to individuals with an obstructive ventilatory defect failed to find an association between glucose or lipid levels and lung function impairment, in line with our present results as well as previous ones [45,46].

The phenotypic heterogeneity of COPD patients and many confounding factors must be considered when comparing the interaction between COPD and metabolic variables across studies. The prevalence of obesity in COPD is highly variable between studies and countries [47]. Some populations showed higher prevalence of obesity [48] with an over-representation

**Table 2. Multivariate linear regression models.**

|  | Effect (%) [#] | p-value |
|---|---|---|
| **Triglycerides** |  |  |
| COPD | +5.5 | 0.262 |
| Age (years) | **-0.6** | **0.029** |
| BMI Kg/m$^2$ | **+3.8** | **<0.001** |
| **Total/HDL cholesterol** |  |  |
| COPD | +1.2 | 0.721 |
| Sex (men) | **+10.3** | **0.005** |
| BMI Kg/m$^2$ | **+2.3** | **<0.001** |
| Hypolipidemic (yes) | **-18.0** | **<0.001** |
| **HOMA-IR** |  |  |
| COPD | -1.8 | 0.858 |
| Sex (men) | **+27.0** | **0.020** |
| BMI Kg/m$^2$ | **+12.0** | **<0.001** |
| **VAT CSA** |  |  |
| COPD | +1.1 | 0.866 |
| Age (years) | **+1.0** | **0.007** |
| Sex (men) | **+14.7** | **0.047** |
| Current smoker (yes) | **-32.6** | **<0.001** |
| Pack-years (n) | **+0.7** | **<0.001** |

Significant p-values are shown in bold.

#: effect on variable in %, per increase in variable. COPD: chronic obstructive pulmonary disease; BMI: body mass index; HDL: High Density Lipoprotein; HOMA-IR: Homeostasis Model Assessment of Insulin Resistance; VAT CSA: Visceral Adipose Tissue Cross-sectionnal Area. Only significant factors and COPD are kept in the model by a backward selection.

in patients with moderate airflow limitation [49,50], whereas in the worldwide population-based BOLD study [47], obesity was less frequent in COPD than in non-COPD. The importance of BMI as a confounding factor in the observed link between COPD and metabolic parameters is clearly illustrated in our data (**S1 Fig**, **Table 2**). In multivariate analyses, BMI was the factor with the strongest association with the metabolic parameters studied. In the same way, treatment with inhaled corticosteroids (present in only 23% of our COPD subjects) could also confound the relationship between COPD, metabolism and adipose tissue accumulation. Inhaled corticosteroids have been related to a 3-fold increase in the likelihood of having a VAT > 75[th] percentile (**S7 Table**). Based on these considerations, it becomes obvious that differences in population phenotypes across studies could at least partially account for inconsistent conclusions about COPD being a risk factor for altered metabolic status [5]. In this regard, data obtained from clinical cohorts are unlikely to be generalizable to the populational level where the majority of patients has only mild to moderate COPD.

Our study has some limitations. First, given the relatively small sample size, a lack of statistical power could be proposed to explain the absence of differences in endpoints between COPD subjects and controls. However, the similitude in the distribution of metabolic variables and obesity in the two groups studied makes this explanation unlikely. Second, the relatively small size of our otherwise well phenotyped sample could have led to a lesser representative image of the population than did the entire CanCOLD cohort. Despite this, the distribution of study participants' characteristics in this sub-study was very similar to that of the entire cohort [51], with a majority of subjects with GOLD 1 and few GOLD 3 and 4 COPD. Furthermore,

only 30% of individuals with COPD in this sub-study were previously diagnosed with the disease, another similitude with other population-based cohorts [52], providing further reassurance regarding how representative the present cohort is of the general population. That said, despite all the care devoted to building a cohort of individuals representative of the general population, some biases may still be present. For example, the most fragile or diseased subjects would probably be less inclined to participate in a clinical study. Third, focusing on a representative and occidental population of COPD, our findings do not necessarily apply to individuals with severe COPD or to those exhibiting particular phenotypes (inflammatory, underweight or obese, with preponderant vascular comorbidities). Finally, physical activity, an important confounder for cardiovascular risk, was not included in the analysis; also, sleep apnea, another potential contributor, was underdiagnosed by far in this cohort when considering the reported prevalence.

## Conclusions

In our cohort randomly drawn from the general population in which individuals with COPD mostly had mild-to-moderate airflow limitation, no difference in the distribution of metabolic parameters appeared compared to control subjects. As such, COPD did not emerge as a specific risk factor for metabolic disorders or visceral adiposity. Although a strong mechanistic rationale can be developed for the existence of physiopathological links between chronic respiratory diseases and dyslipidemia, insulin resistance or visceral adiposity, their existence is likely restricted to specific phenotypes or to the most severely affected patients who are not widely represented in the general population.

## Supporting information

**S1 Table. Multivariate logistic regression on triglycerides > 1.5 mmol/L.**
(DOCX)

**S2 Table. Multivariate logistic regression on triglycerides > 1.5 mmol/L, COPD 2+ only.**
(DOCX)

**S3 Table. Multivariate logistic regression on TC/HDL > 4.**
(DOCX)

**S4 Table. Multivariate logistic regression on TC/HDL > 4, COPD 2+ only.**
(DOCX)

**S5 Table. Multivariate logistic regression on HOMA-IR > 3.**
(DOCX)

**S6 Table. Multivariate logistic regression on HOMA-IR > 3, COPD 2+ only.**
(DOCX)

**S7 Table. Multivariate logistic regression on visceral adipose tissue cross-sectional area (VAT CSA) > 75th percentile by sex of the total population.**
(DOCX)

**S8 Table. Multivariate logistic regression on visceral adipose tissue cross-sectional area (VAT CSA) > 75th percentile by sex of the total population, COPD 2+ only.**
(DOCX)

**S1 Fig. Univariate analysis stratified by BMI for all metabolic parameters.** COPD: chronic obstructive pulmonary disease; BMI: body mass index; HDL: high density lipoprotein; HOMA-IR:

homeostasis model assessment of insulin resistance; VAT CSA: visceral adipose tissue cross-sectionnal Area on L4-L5. $p > 0.05$ for all between-group (COPD vs. controls) comparisons. (TIF)

**S1 Dataset. Anonymised data.**
(XLSX)

## Acknowledgments

We wish to thank the participants of this cohort and support staff who made the study possible. We also thank Gaétan Daigle for his statistical advice, and Dr Yves Deshaies for proofreading English.

CanCOLD Collaborative research Group:

Executive Committee: Jean Bourbeau, lead author (McGill University, Montreal, QC, Canada, jean.bourbeau@mcgill.ca); Wan C. Tan, J. Mark FitzGerald, D. D. Sin (UBC, Vancouver, BC, Canada); D. D. Marciniuk (University of Saskatoon, Saskatoon, SASK, Canada) D. E. O'Donnell (Queen's University, Kingston, ON, Canada); Paul Hernandez (University of Halifax, Halifax, NS, Canada); Kenneth R. Chapman (University of Toronto, Toronto, ON, Canada); Robert Cowie (University of Calgary, Calgary, AB, Canada); Shawn Aaron (University of Ottawa, Ottawa, ON, Canada); F. Maltais (University of Laval, Quebec City, QC, Canada); International Advisory Board: Jonathon Samet (the Keck School of Medicine of USC, CA, USA); Milo Puhan (John Hopkins School of Public Health, Baltimore, USA); Qutayba Hamid (McGill University, Montreal, QC, Canada); James C. Hogg (UBC James Hogg Research Center, Vancouver, BC, Canada). Operations Center: Jean Bourbeau (Principal Investigator), Carole Baglole, Carole Jabet, Palmina Mancino, Yvan Fortier (University of McGill, Montreal, QC, Canada); Wan C. Tan (co-PI), Don Sin, Sheena Tam, Jeremy Road, Joe Comeau, Adrian Png, Harvey Coxson, Miranda Kirby, Jonathon Leipsic, Cameron Hague (University of British Columbia James Hogg Research Center, Vancouver, BC, Canada). Economic Core: Mohsen Sadatsafavi (University of British Columbia, Vancouver, BC). Public Health Core: Teresa To, Andrea Gershon (University of Toronto). Data Management and Quality Control: Wan C. Tan, Harvey Coxson (UBC, Vancouver, BC, Canada); Jean Bourbeau, Pei-Zhi Li, Jean-Francois Duquette, Yvan Fortier, Andrea Benedetti, Denis Jensen (McGill University, Montreal, QC, Canada), Denis O'Donnell (Queen's University, Kingston, ON, Canada). Field Centres: Wan C. Tan (PI), Christine Lo, Sarah Cheng, Cindy Fung, Nancy Ferguson, Nancy Haynes, Junior Chuang, Licong Li, Selva Bayat, Amanda Wong, Zoe Alavi, Catherine Peng, Bin Zhao, Nathalie Scott-Hsiung, Tasha Nadirshaw (UBC James Hogg Research Center, Vancouver, BC, Canada); Jean Bourbeau (PI), Palmina Mancino, David Latreille, Jacinthe Baril, Laura Labonte (McGill University, Montreal, QC, Canada); Kenneth Chapman (PI), Patricia McClean, Nadeen Audisho (University of Toronto, Toronto, ON, Canada); Brandie Walker, Robert Cowie (PI), Ann Cowie, Curtis Dumonceaux, Lisette Machado(University of Calgary, Calgary, AB, Canada); Paul Hernandez (PI), Scott Fulton, Kristen Osterling (University of Halifax, Halifax, NS, Canada); Shawn Aaron (PI), Kathy Vandemheen, Gay Pratt, Amanda Bergeron (University of Ottawa, Ottawa, ON, Canada); Denis O'Donnell (PI), Matthew McNeil, Kate Whelan (Queen's University, Kingston, ON, Canada); Francois Maltais (PI), Cynthia Brouillard (University of Laval, Quebec City, QC, Canada); Darcy Marciniuk (PI), Ron Clemens, Janet Baran (University of Saskatoon, Saskatoon, SK, Canada).

## Author Contributions

**Conceptualization:** Damien Viglino.

**Data curation:** Damien Viglino, Mickaël Martin, François Maltais.

**Formal analysis:** Damien Viglino, François Maltais.

**Funding acquisition:** François Maltais.

**Investigation:** Mickaël Martin, Cynthia Brouillard, Jean-Pierre Després, Natalie Alméras, Wan C. Tan, Valérie Coats, Jean Bourbeau, François Maltais.

**Methodology:** Damien Viglino, Marie-Eve Piché, François Maltais.

**Resources:** Jean Bourbeau, François Maltais.

**Supervision:** François Maltais.

**Validation:** Marie-Eve Piché, Jean-Pierre Després.

**Visualization:** Damien Viglino.

**Writing – original draft:** Damien Viglino.

**Writing – review & editing:** Mickaël Martin, Marie-Eve Piché, Jean-Pierre Després, Natalie Alméras, Wan C. Tan, Valérie Coats, Jean Bourbeau, Jean-Louis Pépin, François Maltais.

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
