## [Decision Letter · Decision Letter 0]

18 Feb 2020

PONE-D-19-33841

Metabolic profiles among COPD and controls in the CanCOLD population-based cohort

PLOS ONE

Dear Dr VIGLINO,

Thank you for submitting your manuscript to PLOS ONE. After careful consideration, we feel that it has merit but does not fully meet PLOS ONE’s publication criteria as it currently stands. Therefore, we invite you to submit a revised version of the manuscript that addresses the points raised during the review process.

Please pay special attention to the concerns in the methods, which are primarily about sample size, representation, diagnosis, and result analysis.

We would appreciate receiving your revised manuscript by Apr 03 2020 11:59PM. To enhance the reproducibility of your results, we recommend that if applicable you deposit your laboratory protocols in protocols.io, where a protocol can be assigned its own identifier (DOI) such that it can be cited independently in the future. For instructions see: http://journals.plos.org/plosone/s/submission-guidelines#loc-laboratory-protocols

We look forward to receiving your revised manuscript.

Kind regards,

Qinghua Sun, MD, PhD

Academic Editor

PLOS ONE

Journal Requirements:

2.. Thank you for including your ethics statement:

'All patients provided written informed consent and the study were approved by ethics

committees of participating centres. CanCOLD (ClinicalTrials.gov: NCT00920348)

steering and scientific committees approved the sub study protocol'

3. Please provide additional details regarding healthy participants consent. In the ethics statement in the Methods and online submission information, please ensure that you have specified (1) whether consent was suitably informed and (2) what type you obtained (for instance, written or verbal). If your study included minors under age 18, state whether you obtained consent from parents or guardians. If the need for consent was waived by the ethics committee, please include this information.

4. We noticed you have some minor occurrence(s) of overlapping text with the following previous publication(s), which needs to be addressed:

https://doi.org/10.2147/COPD.S168963

https://doi.org/10.1152/ajpendo.00505.2018

In your revision ensure you cite all your sources (including your own works), and quote or rephrase any duplicated text outside the Methods section. Further consideration is dependent on these concerns being addressed.

5. One of the noted authors is a group or consortium CanCOLD Collaborative Research Group and the Canadian Respiratory Research Network. In addition to naming the author group, please list the individual authors and affiliations within this group in the acknowledgments section of your manuscript. Please also indicate clearly a lead author for this group along with a contact email address.

6.

Thank you for stating the following in the Competing Interests section:

"The authors have declared that no competing interests exist"

We note that you received funding from a commercial source: Astra Zeneca Canada Ltd; Boehringer Ingelheim Canada Ltd; GlaxoSmithKline Canada Ltd; Novartis; Almirall; Merck Nycomed; Pfizer Canada Ltd; and Theratechnologies.

Reviewers' comments:

Reviewer's Responses to Questions

**Comments to the Author**

1. Is the manuscript technically sound, and do the data support the conclusions?

Reviewer #1: Yes

Reviewer #2: Partly

Reviewer #3: Partly

2. Has the statistical analysis been performed appropriately and rigorously? 

Reviewer #1: Yes

Reviewer #2: No

Reviewer #3: I Don't Know

3. Have the authors made all data underlying the findings in their manuscript fully available?

Reviewer #1: Yes

Reviewer #2: Yes

Reviewer #3: Yes

4. Is the manuscript presented in an intelligible fashion and written in standard English?

Reviewer #1: Yes

Reviewer #2: Yes

Reviewer #3: Yes

5. Review Comments to the Author

Reviewer #1: This article tries to find out if COPD contributes independently to development of some cardiometabolic risk factors (visceral adipose tissue, assessment of insulin resistance: HOMA-IR and lipids profile).

Major points:

1) The study was with random population sampling, and at last it included 263 participants. As indicated in the limitations, it could be a small sample size. Is this sample representative of the general population? Why did you do only a sub-analysis restricted to individuals with an obstructive ventilatory defect?

2) In the results and discussion section, we check that univariate and multivariate analyses failed to show COPD as a predictor of metabolic disorders and visceral adiposity in the cohort. Some confounding factors like BMI and different phenotype of COPD could explain the conclusions, what more confounding factors may be related to these results? Have you found any article where it appears?

Minor points:

1) In the discussion section, line 23: Leone et al. instead of Leone and collegues.

Reviewer #2: Viglino and colleagues report that they found no differences in the distribution of metabolic parameters among patients with and without COPD in a Canadian cohort. I would like to address some concerns.

Major comments:

- The sample size is very small (263 subjects including 119 controls), also in comparison to numerous other studies on the same subject: In a recent meta-analysis (Cebron Lipovec, COPD 2016) the majority of 19 included studies had more patients than the present study. Authors should explain why the majority of CanCOLD participants were not included in the present sub-analysis and how the sub-analysis group was chosen.

- Authors state several times that they wanted to investigate whether COPD “alters” or “predicts” the metabolic risk profile. This cannot be shown in a cross sectional analysis, but must be investigated in longitudinal and/or intervention studies. On the other hand, authors report on “prospective classification” of COPD and on a “follow-up investigation”. These longitudinal data should be shown.

- Severe forms of COPD (GOLD III and IV) are clearly underrepresented. Does this mirror the severity of the total CanCOLD population? Also, additionally to the GOLD classification the symptoms and the risk of exacerbations (Group A-D of the GOLD Guideline 2011) should be taken into account.

Minor comments:

- Authors state several times that the chosen cohort would be representative for the general population. How was this defined and tested? Numbers vary, but epidemiologist normally demand around 40% participants of a population to call a sample “representative for the whole population”.

- Abbreviations throughout the abstract and the text should be explained at first mention (e.g. “L4-L5” in the abstract).

- Some of the variables mentioned in the baseline characteristics are probably normally distributed (age, body-mass index, waist-hip ratio). These should be presented with mean and standard deviation instead of median and interquartile range.

- Authors should comment on the low percentage of patients taking inhalative short- and long-acting bronchodilators.

Reviewer #3: 1.It would have been better if the spirometry data were included in six standard items including: FEV1-FVC-FEV1/FVC-PEF-FEF25-75 and VEXt or Evol.

2. How is sleep apnea measured?

3.Please consult with an epidemiologist for statistical analysis.

4.Please edit minor changes in English grammar.

6. PLOS authors have the option to publish the peer review history of their article (what does this mean?). If published, this will include your full peer review and any attached files.

Reviewer #1: No

Reviewer #2: No

Reviewer #3: No

---

## [Author Response · Author response to Decision Letter 0]

3 Mar 2020

2nd March, 2020

Dear Editor, 

Please find enclosed the revised version of our article entitled “Metabolic profiles among COPD and controls in the CanCOLD population-based cohort”.

We are grateful to the reviewers for their time and effort in evaluating our manuscript, and appreciate their comments. The following is a point-by-point response to the reviewers’ comments and journal requirements.

Journal Requirements:

Response: This has been done accordingly

'All study participants provided written informed consent and the study was approved by the ethics committees of participating centres. CanCOLD(ClinicalTrials.gov:NCT00920348)

steering and scientific committees approved the sub study protocol'

Response: This has been done accordingly:

Manuscript, Methods: “The study was approved by the local ethics committee (Comité d’éthique du centre de recherche de l’Institut Universitaire de Cardiologie et de Pneumologie de Québec, IRB N° 20690, Study N° 2012-1359). CanCOLD (ClinicalTrials.gov: NCT00920348) steering and scientific committees approved the sub-study protocol. All study participants signed written consent before inclusion.”

3. Please provide additional details regarding healthy participants consent. In the ethics statement in the Methods and online submission information, please ensure that you have specified (1) whether consent was suitably informed and (2) what type you obtained (for instance, written or verbal). If your study included minors under age 18, state whether you obtained consent from parents or guardians. If the need for consent was waived by the ethics committee, please include this information.

Response: All study participants, controls and COPD, signed written consent. The following statement has been added:

Manuscript, Methods: “All study participants signed written consent before inclusion.”

4. We noticed you have some minor occurrence(s) of overlapping text with the following previous publication(s), which needs to be addressed:

https://doi.org/10.2147/COPD.S168963

https://doi.org/10.1152/ajpendo.00505.2018

In your revision ensure you cite all your sources (including your own works), and quote or rephrase any duplicated text outside the Methods section. Further consideration is dependent on these concerns being addressed.

Response: We supposed that the overlap part is the description of the visceral adipose tissue assessment in the Methods section. We used the same text as we did in previous publications since this technique is strictly identical and performed by the same team with the same tools in our study and in both previous publications cited :

“Abdominal fat distribution was assessed with L4-L5 CT scans, and images were analyzed without knowledge of the clinical status of the subjects using a specialized software (Tomovision SliceOMatic 4.3 Rev-6f software, Montreal, Quebec, Canada). The detailed method used for the images analysis has been reported [31]: visceral adipose tissue (VAT) area was determined by delineating the middle of the muscle wall surrounding the abdominal cavity.”

We now quote both works after this part, and have made reformulation efforts to avoid strict overlap.

5. One of the noted authors is a group or consortium CanCOLD Collaborative Research Group and the Canadian Respiratory Research Network. In addition to naming the author group, please list the individual authors and affiliations within this group in the acknowledgments section of your manuscript. Please also indicate clearly a lead author for this group along with a contact email address.

Response: This has been done accordingly

6. Thank you for stating the following in the Competing Interests section:

"The authors have declared that no competing interests exist"

We note that you received funding from a commercial source: Astra Zeneca Canada Ltd; Boehringer Ingelheim Canada Ltd; GlaxoSmithKline Canada Ltd; Novartis; Almirall; Merck Nycomed; Pfizer Canada Ltd; and Theratechnologies.

Response: All funding sources (study, CanCOLD cohort) were properly cited in the “Financial disclosure” statement during the submission process. Indeed, we did not mention the individual competing interest outside the present work. 

Our amended and detailed Competing Interests Statement is now available in the cover letter as requested.

 

Reviewers' comments:

Reviewer #1: 

This article tries to find out if COPD contributes independently to development of some cardiometabolic risk factors (visceral adipose tissue, assessment of insulin resistance: HOMA-IR and lipids profile).

Major points:

1) The study was with random population sampling, and at last it included 263 participants. As indicated in the limitations, it could be a small sample size. Is this sample representative of the general population? Why did you do only a sub-analysis restricted to individuals with an obstructive ventilatory defect?

Response: This CanCOLD sub-study has been specifically designed to evaluate whether COPD is a predictor of cardiometabolic health and to document if the determinants of cardiometabolic health are similar in COPD and in healthy subjects with normal lung function, excluding those other spirometric abnormalities such as a restrictive pattern. Some non-COPD subjects with abnormal but non-obstructive spirometric pattern were excluded. The substudy involved numerous additional examinations and follow-up, including abdominal CT scans and biological samples, which could not be obtained in all CanCOLD participating centres. Indeed, all CanCOLD participants in two CanCOPD participating centres were involved. Despite a smaller sample size than with the entire CanCOLD cohort, we expect from the random population sampling method, as opposed to convenience sampling, a cohort of COPD subjects (even non-diagnosed previously) and healthy subjects that is representative of the general population. Furthermore, the distribution of subjects characteristics in this sub-study is very similar to that of the entire CanCOLD cohort (see Labonté et al, AJRCCM 2016 194(3), 285-298), with a majority of GOLD 1 grades, few GOLD 3 or 4, and only 30% of COPD patients with previously diagnosed COPD.

Indeed, these limitations are now clearly stated in the manuscript : 

“given the relatively small sample size, a lack of statistical power could be proposed to explain the absence of differences in endpoints between COPD subjects and controls. However, the similitude in the distribution of metabolic variables and obesity in the two groups studied makes this explanation unlikely. Second, the relatively small size of our otherwise well phenotyped sample could have led to a lesser representative image of the population than did the entire CanCOLD cohort. Despite this, the distribution of study participants’ characteristics in this sub-study was very similar to that of the entire cohort [51], with a majority of subjects with GOLD 1 and few GOLD 3 and 4 COPD. Furthermore, only 30% of individuals with COPD in this sub-study were previously diagnosed with the disease, another similitude with other population-based cohorts [52], providing further reassurance regarding how representative the present cohort is of the general population. That said, despite all the care devoted to building a cohort of individuals representative of the general population, some biases may still be present. For example, the most fragile or diseased subjects would probably be less inclined to participate in a clinical study.”

2) In the results and discussion section, we check that univariate and multivariate analyses failed to show COPD as a predictor of metabolic disorders and visceral adiposity in the cohort. Some confounding factors like BMI and different phenotype of COPD could explain the conclusions, what more confounding factors may be related to these results? Have you found any article where it appears?

Response: (“Failed” was used in the manuscript with the meaning of “negative”, not “failure”). More than explaining this conclusion, taking into account BMI and other factors related to metabolic parameters shows that COPD status per se does not seem to modify metabolic parameters. This means for us that COPD patients may have different morphological characteristics, but at identical morphological characteristics, the metabolic parameters are not modified by the presence of COPD. Furthermore, our results are consistent even in univariate analysis.

To our knowledge, the main known confounding factors of cardiometablic health were taken into account in our analyses, and were sufficient to correct an effect which is not due to COPD. Adding functional status or sedentarity scores in models (links with metabolic syndrome well described in the literature) could have been considered, but no such prospective evaluation was planned.

Minor points:

1) In the discussion section, line 23: Leone et al. instead of Leone and colleagues.

Response: This has been corrected.

Reviewer #2: 

Viglino and colleagues report that they found no differences in the distribution of metabolic parameters among patients with and without COPD in a Canadian cohort. I would like to address some concerns.

Major comments:

1) The sample size is very small (263 subjects including 119 controls), also in comparison to numerous other studies on the same subject: In a recent meta-analysis (Cebron Lipovec, COPD 2016) the majority of 19 included studies had more patients than the present study. Authors should explain why the majority of CanCOLD participants were not included in the present sub-analysis and how the sub-analysis group was chosen.

And:

3) Severe forms of COPD (GOLD III and IV) are clearly underrepresented. Does this mirror the severity of the total CanCOLD population? Also, additionally to the GOLD classification the symptoms and the risk of exacerbations (Group A-D of the GOLD Guideline 2011) should be taken into account.

Response: This sub-study has been designed specifically evaluate whether COPD is a predictor of cardiometabolic health and to document if the determinants of cardiometabolic health are similar in COPD and in healthy subjects with normal lung function, excluding those other spirometric abnormalities such as a restrictive pattern. 

The substudy involved numerous additional examinations and follow-up, including abdominal CT scans and biological samples, which could not be obtained in all CanCOLD participating centres. Indeed, all CanCOLD participants in two CanCOPD participating centres were involved. Despite a smaller sample size than with the entire CanCOLD cohort, we expect from the random population sampling method, as opposed to convenience sampling, a cohort of COPD subjects (even non-diagnosed previously) and healthy subjects that is representative of the general population. Furthermore, the distribution of subjects characteristics in this sub-study is very similar to that of the entire CanCOLD cohort (see Labonté et al, AJRCCM 2016 194(3), 285-298), with a majority of GOLD 1 grades, few GOLD 3 or 4, and only 30% of COPD patients with previously diagnosed COPD.

These limitations are now clearly stated in the manuscript : 

“given the relatively small sample size, a lack of statistical power could be proposed to explain the absence of differences in endpoints between COPD subjects and controls. However, the similitude in the distribution of metabolic variables and obesity in the two groups studied makes this explanation unlikely. Second, the relatively small size of our otherwise well phenotyped sample could have led to a lesser representative image of the population than did the entire CanCOLD cohort. Despite this, the distribution of study participants’ characteristics in this sub-study was very similar to that of the entire cohort [51], with a majority of subjects with GOLD 1 and few GOLD 3 and 4 COPD. Furthermore, only 30% of individuals with COPD in this sub-study were previously diagnosed with the disease, another similitude with other population-based cohorts [52], providing further reassurance regarding how representative the present cohort is of the general population. That said, despite all the care devoted to building a cohort of individuals representative of the general population, some biases may still be present. For example, the most fragile or diseased subjects would probably be less inclined to participate in a clinical study.”

Regarding the A,B,C,D GOLD classification, the proportion of patients has been added to the table 1. The CAT scores and the A,B,C,D classification have been added in the database (supporting information).

We tested a subgroup analysis using the A,B,C, D GOLD classification instead of the dichotomy GOLD 1 vs. GOLD 2+, without modification of the results obtained. This similitude between the two analyses is likely due to the small number of patients in categories C (2.1%) and D (6.9%). For the sake of clarity of the message, we have not included these analyzes in the results. 

2) Authors state several times that they wanted to investigate whether COPD “alters” or “predicts” the metabolic risk profile. This cannot be shown in a cross sectional analysis, but must be investigated in longitudinal and/or intervention studies. On the other hand, authors report on “prospective classification” of COPD and on a “follow-up investigation”. These longitudinal data should be shown.

Response: We agree that the vocabulary that was used to illustrate our interpretation of the findings was not totally clear. First, the “metabolic risk profile” is commonly admitted to describe risk factors related to metabolic disorders, not the occurrence of events.

-Prospective classification : Most studies include previously diagnosed COPD patients. The classification into COPD or Healthy was done prospectively in the CanCOLD cohort since the clinical status of the individuals was unknown at the time of study inclusion. This study design was used to build a cohort of subjects who would be representative of the general population, also capturing non-diagnosed patients. The purpose of this report was to report the cross-sectional data and not the longitudinal follow-up which is still incomplete. We agree with the reviewer that only longitudinal data could be used to infer causal relationship between the presence of COPD and cardiometabolic health. We have amended the text of the revised manuscript accordingly. 

-Predicts : Although, it is possible by using mathematical models to “predict” a variable or characteristic from other characteristics of a subject at a given time, we have we changed some formulations to avoid misunderstandings :

Introduction : “In the present follow-up investigation […]”

Results : “In multivariate analysis, the COPD status (or COPD 2+) did not emerge as predictors was not associated with triglyceride >1.5 mmol/L […]”

Discussion : “Univariate and multivariate analyses failed to show COPD as a predictor of showed an absence of association between COPD and metabolic disorders or visceral adiposity.”

3) Authors state several times that the chosen cohort would be representative for the general population. How was this defined and tested? Numbers vary, but epidemiologist normally demand around 40% participants of a population to call a sample “representative for the whole population”.

Response: It would be extremely difficult and costly to support a study with this level of phenotyping involving 40% of its initial population. The remaining alternative is therefore random sampling, which is a unique contribution of CanCOLD. We discuss within the limits of the study a possible lack of power, but it is still difficult to question the representativeness of a random sample, especially since the characteristics of the patients are similar to those of the entire cohort (see above).

4) Abbreviations throughout the abstract and the text should be explained at first mention (e.g. “L4-L5” in the abstract).

Response: “4th/5th lumbar vertebrae level” has been added to the first mention of L4-L5 in the manuscript.

5) Some of the variables mentioned in the baseline characteristics are probably normally distributed (age, body-mass index, waist-hip ratio). These should be presented with mean and standard deviation instead of median and interquartile range.

Response: Only the waist-hip ratio and PEF were normally distributed in both groups (COPD and Controls) using the Kolmogorov-Smirnov test, and the same result is obtained using the Shapiro-Wilk test. 

These variables are now presented with mean and 95% CI, the Table legend and the “data analysis” section have been modified accordingly.

6) Authors should comment on the low percentage of patients taking inhalative short- and long-acting bronchodilators.

Response: As explained above, only 30% of CanCOLD COPD patients were diagnosed with the disease before the study. This point has been added in the discussion :

“Furthermore, only 30% of individuals with COPD in this sub-study were previously diagnosed with the disease, another similitude with other population-based cohorts [52], providing further reassurance regarding how representative the present cohort is of the general population. That said, despite all the care devoted to building a cohort of individuals representative of the general population, some biases may still be present. For example, the most fragile or diseased subjects would probably be less inclined to participate in a clinical study.”

Reviewer #3: 

1) It would have been better if the spirometry data were included in six standard items including: FEV1-FVC-FEV1/FVC-PEF-FEF25-75 and VEXt or Evol.

Response: PEF and FEF 25-75 have been added in the Table 1 accordingly, and in the database (supporting information). 

The Vext being a measure of quality of the test only, it is very unusual to present it, all the spirometry being made according to standards of quality, with a new test if the value of the Vext was abnormal. Furthermore, the quality and the validity of the spirometric procedures in CanCOLD have been validated (Tan WC et al. COPD 2014;11:143-151)

2) How is sleep apnea measured?

Response: As explained in the limitations, no systematic polysomnography was planned in the cohort (“sleep apnea another potential contributor is by far underdiagnosed in this cohort when looking at the reported prevalence.”). Diagnosed sleep apnea was searched through standardized questionnaires, including the Pittsburg Sleep Quality Index (see Shorofsky M et al. Chest 2019;156:852-863, “Impaired Sleep Quality in COPD Is Associated With Exacerbations: The CanCOLD Cohort Study”) and the presence of specific treatments (CPAP). Indeed, since the cohort aimed to include a representative population, it also had to include undiagnosed patients (COPD, sleep apnea), and a polysomnography would have created an intervention bias. The low proportion of sleep apnea diagnosed is representative of the general population.

We cited more clearly this point in the manuscript :

Methods : “Although no sleep studies were done in CanCOLD, the presence of sleep apnea was documented based on the use of continuous airway positive pressure (CPAP) and on standardized questionnaires, including the Pittsburg Sleep Quality Index [31].”

3) Please consult with an epidemiologist for statistical analysis.

Response: We do not know if your comment concerns the editor (request for statistical revision) or our team. J. Bourbeau and D. Viglino are qualified epidemiologists, and the statistical analysis plan has been reviewed by Dr Gaétan Daigle, a professional statistician at Laval University, QC, CA.

4) Please edit minor changes in English grammar.

Response: This version of the manuscript has been proofread for English. In the absence of specific points to help us, we hope the changes you pointed out have been made.

---

## [Decision Letter · Decision Letter 1]

17 Mar 2020

Metabolic profiles among COPD and controls in the CanCOLD population-based cohort

PONE-D-19-33841R1

Dear Dr. VIGLINO,

We are pleased to inform you that your manuscript has been judged scientifically suitable for publication and will be formally accepted for publication once it complies with all outstanding technical requirements.

With kind regards,

Qinghua Sun, MD, PhD

Academic Editor

PLOS ONE

Additional Editor Comments (optional):

Reviewers' comments:

Reviewer's Responses to Questions

**Comments to the Author**

1. If the authors have adequately addressed your comments raised in a previous round of review and you feel that this manuscript is now acceptable for publication, you may indicate that here to bypass the “Comments to the Author” section, enter your conflict of interest statement in the “Confidential to Editor” section, and submit your "Accept" recommendation.

Reviewer #2: All comments have been addressed

Reviewer #3: All comments have been addressed

2. Is the manuscript technically sound, and do the data support the conclusions?

Reviewer #2: Yes

Reviewer #3: Yes

3. Has the statistical analysis been performed appropriately and rigorously? 

Reviewer #2: Yes

Reviewer #3: Yes

4. Have the authors made all data underlying the findings in their manuscript fully available?

Reviewer #2: Yes

Reviewer #3: Yes

5. Is the manuscript presented in an intelligible fashion and written in standard English?

Reviewer #2: Yes

Reviewer #3: Yes

6. Review Comments to the Author

Reviewer #2: (No Response)

Reviewer #3: I do not any additional comments for the author, including concerns about dual publication, research ethics, or publication ethics.

Thank you for author's responses.

7. PLOS authors have the option to publish the peer review history of their article (what does this mean?). If published, this will include your full peer review and any attached files.

Reviewer #2: No

Reviewer #3: No

---

## [Editor Report · Acceptance letter]

26 Mar 2020

PONE-D-19-33841R1 

Metabolic profiles among COPD and controls in the CanCOLD population-based cohort 

Dear Dr. VIGLINO:

I am pleased to inform you that your manuscript has been deemed suitable for publication in PLOS ONE. Congratulations! Your manuscript is now with our production department. 

With kind regards,

on behalf of

Dr Qinghua Sun 

Academic Editor

PLOS ONE